# Non-Contact Monitoring of Human Vital Signs Using FMCW Millimeter Wave Radar in the 120 GHz Band

**DOI:** 10.3390/s21082732

**Published:** 2021-04-13

**Authors:** Wenjie Lv, Wangdong He, Xipeng Lin, Jungang Miao

**Affiliations:** 1School of Electronic Information Engineering, Beihang University, Beijing 100191, China; hwd19930315@buaa.edu.cn (W.H.); zy1802132@buaa.edu.cn (X.L.); jmiaobremen@buaa.edu.cn (J.M.); 2Key Laboratory of Microwave Perception & Safety Application, Beihang University, Beijing 100191, China

**Keywords:** non-contact measurement, millimeter wave radar, narrow beam, adaptive notch, frequency correction, fast independent component analysis

## Abstract

A non-contact heartbeat/respiratory rate monitoring system was designed using narrow beam millimeter wave radar. Equipped with a special low sidelobe and small-sized antenna lens at the front end of the receiving and transmitting antennas in the 120 GHz band of frequency-modulated continuous-wave (FMCW) system, this sensor system realizes the narrow beam control of radar, reduces the interference caused by the reflection of other objects in the measurement background, improves the signal-to-clutter ratio (SCR) of the intermediate frequency signal (IF), and reduces the complexity of the subsequent signal processing. In order to solve the problem that the accuracy of heart rate is easy to be interfered with by respiratory harmonics, an adaptive notch filter was applied to filter respiratory harmonics. Meanwhile, the heart rate obtained by fast Fourier transform (FFT) was modified by using the ratio of adjacent elements, which helped to improve the accuracy of heart rate detection. The experimental results show that when the monitoring system is 1 m away from the human body, the probability of respiratory rate detection error within ±2 times for eight volunteers can reach 90.48%, and the detection accuracy of the heart rate can reach 90.54%. Finally, short-term heart rate measurement was realized by means of improved empirical mode decomposition and fast independent component analysis algorithm.

## 1. Introduction

With the development of medical technology and the continuous enhancement of people’s health consciousness, the measurement device which can obtain human physiological information anytime and anywhere has been brought to the fore in recent years. It is known to us all that respiration and heartbeat are important vital signs of the human body, and whether their frequency is normal or not is an important basis for judging the changes of patients’ health status. Traditional devices for vital signs monitoring usually adopt contact monitoring methods, which are designed mainly to meet the needs of clinical medical monitoring and disease diagnosis. All of those shortcomings such as high accuracy requirements, great bulk, high cost, complex operation, uncomfortable body feeling, being not suitable for long-term monitoring and the risk of cross infection work together to limit their applications and make it impossible for them to be popularized to individuals and families [1]. Therefore, the technology of reliable and real-time monitoring of cardiopulmonary parameters without contact is of great significance in the fields of daily health monitoring, special population monitoring, medical diagnosis, disaster relief and security protection [2,3]. A non-contact vital signs monitoring system has the following advantages: (1) it is simple to use and less limited by monitoring conditions; reliable heart/respiratory rate monitoring can be performed anytime, anywhere; (2) it can meet the needs of long-term signals detection and the discomfort caused by the signal detection process can be minimized.

As a kind of high-sensitivity detection equipment, millimeter wave radar can be used to monitor weak physiological signals [4,5]. Compared with other vital signs sensors, the millimeter wave radar sensor based on the Doppler effect has the advantages of portability, non-contact, high sensitivity and not being affected by environmental factors. In most frequency bands of microwaves and millimeter waves, the attenuation of electromagnetic waves is negligible when penetrating clothes, bags and other materials. These characteristics of microwave and millimeter wave make it very suitable for non-contact detection of human vital signs through clothing. In addition, the wavelength of the millimeter wave can be compared with the displacement caused by human physiological movements, while the phase modulation of reflected waves caused by physiological activities are inversely proportional to the wavelength. As a result, millimeter wave radar working at a higher frequency can detect the human body more precisely without requiring a larger antenna diameter, which is very beneficial to the miniaturization and portability of the equipment. In order to realize the non-contact detection of vital signs, researchers used the Doppler principle to extract the phase change information caused by the weak physiological movements on the body surface from the echo of radar wave, and extracted heartbeat and respiratory signals by means of various signal processing methods, thus developing various non-contact life parameters detection systems [6,7,8,9,10,11].

However, at the present stage, there still exist some problems in the non-contact vital signs monitoring device based on the Doppler radar principle, which can be described as follows: (1) low frequency band microwave radar is widely used, which is featured with wide beam and high radiation power. It is easy to be interfered with by multipath effect and environmental noises in wards with narrow space, which makes the subsequent signal processing algorithm complicated and is not conducive to the extraction of heartbeat/respiratory signals; (2) the traditional signal processing algorithm uses finite impulse response (FIR) filter to process the intermediate frequency (IF) signal phase sequence to realize the separation of breath signals and heartbeat signals. Calculating the signal frequency by means of fast Fourier transform (FFT) cannot solve the problem of respiratory signal harmonic and clutter interferences [12], thus affecting the accurate measurement of heartbeat signal frequency; (3) the FFT algorithm can only calculate the average heart rate over a period of time; its estimation accuracy for short-term heart rate is low. As a result, it cannot judge abnormal physiological phenomena such as uneven heart rate, so its application value in medicine is limited.

On the basis of what has been analyzed above, this paper presents a study on a kind of non-contact measuring device based on narrow beam millimeter radar for human respiratory/heart rate detection and the corresponding measuring methods for adopting frequency-modulated continuous-wave (FMCW) radar in the 120 GHz band. Consisting of FMCW narrow beam millimeter wave radar, analog to digital (AD) acquisition module and upper computer signal processing program, this sensor system possesses obvious advantages over continuous-wave (CW) biological radar, such as no need for calibration, simple operation and high measurement accuracy. The main contribution of this paper is that it improves the signal-to-clutter ratio (SCR) of radar IF signals by using narrow beam lens antenna in vital signs monitoring radar. Meanwhile, in order to meet the accuracy requirements of heart rate measurement, before using FFT to estimate the respiratory and heart rate, respiratory harmonics are filtered out by means of an adaptive notch filter. Then, on the spectrum obtained by FFT, the ratio of absolute values of elements on both sides of the peak heart rate is used to correct the heart rate, thus realizing the high-precision estimation of heart rate. Furthermore, in this paper the algorithm combining complete ensemble empirical mode decomposition with adaptive noise (CEEMDAN) and fast independent component analysis (Fast-ICA) was adopted to study the short-term heart rate measurement technology. The main contents of this paper are as follows: Section 2 introduces the radar system and its working principle, including the hardware design of the narrow beam lens antenna and FMCW millimeter wave radar. Section 3 describes the experimental results in a cramped room. Section 4 summarizes the whole paper and comes to some conclusions.

## 2. Theory and Methods

### 2.1. Principle of Non-Contact Vital Signs Monitoring with FMCW Radar

Physiologically, cardiopulmonary activities during heart beating and breathing can cause chest wall motions, which is the reason why bio-radar can detect cardiopulmonary activities. It was found from previous studies that the maximum relative displacement of the chest wall caused by heart movements have reached 0.6 mm, and the displacement caused by the breathing process was over 12 mm, which has reached the detection conditions of the Doppler radar. Generally speaking, root mean square (RMS) motion is defined as the root mean square value of the motion amplitude of the thoracic surface in the radiation direction of radar antenna. According to specific bio-radar experimental data, the RMS motions of heartbeat and respiratory movements are estimated to be 0.3 mm and 2 mm, respectively; the motion area of the heartbeat is about 10 cm^2^, and that of respiration is about 50 cm^2^ [13]. These estimates are made based on the assumption that the useful vibration signals produced by respiration and heartbeat are linearly superimposed.

The millimeter wave radar in the 120 GHz band is featured with short wavelength and small-sized antenna; narrow beam control can be realized by adding an antenna lens, which is ideal for heartbeat, breath and jitter monitoring based on point displacement measurement. For one thing, the higher the frequency is, the greater the bandwidth that can be provided will be; and according to the range resolution formula Res = c/2B, the range resolution of radar will be higher, which is more beneficial to separating different areas of the body surface and judging whether the body has micro motion. For another, the lens antenna reduces the 3 dB width of the transmitted beam, effectively suppresses the spatial clutter and improves the SCR of IF signals. Meanwhile, when the antenna beam is perpendicular to the chest and heart of the human body, the main beam can concentrate on the small area near the heart of the body surface. Through the displacement-phase coupling effect, the reflected wave can carry the surface change characteristics caused by a strong heartbeat and respiration.

As for the measurement process of human heart and lung parameters, the radar transmits narrow beam FMCW pulse to the human heart, and the reflected echo is received by the receiving antenna and mixed with the local oscillator to obtain the IF output signal. The IF signal is then sampled by high-speed AD, and the composite signals of heartbeat and breath can be obtained by demodulating the central frequency phase of the sampled digital signal. In FMCW radar, the frequency of the transmitted signal varies linearly with time. The transmission signal *s*(*t*) can be expressed as [14]:(1)$$s(t)=e_{t}\cos(2\pi f_{c}t+\pi \frac{B}{T}t^{2}+\phi_{1})+n_{1}(t),$$
where *f_c_* stands for the starting frequency, *B* the signal sweeping bandwidth, and *T* the signal duration. The received signal *r*(*t*) can be regarded as the delay of the transmitted signal, which can be expressed as:(2)$$r(t)=e_{r}\cos(2\pi f_{c}(t-t_{d})+\pi \frac{B}{T}{\left(t-t_{d}\right)}^{2}+\phi_{2})+n_{2}(t),$$
where $t_{d}=2R/c$ represents the interval between the transmission and reception of a signal for a human target with a given distance of *R*. The IF signal output after mixing the local oscillator signal and the received signal can be approximately expressed as follows:(3)$$s(t)*r(t)\propto A\cos(4\pi \frac{BR}{cT}t+\frac{4\pi }{\lambda }R+\phi_{3})+n_{3}(t)=A\cos(4\pi \frac{BR}{cT}t+\varphi_{b}+\phi_{3})+n_{3}(t),$$
where ϕ stands for the initial phase and *n*(*t*) the noise.

In fact, the quadratic term whose frequency is 2*f_c_* + *f* is also included in the above formula, which cannot be detected by conventional instruments and thus can be ignored. As a consequence, the IF signal after mixing carries the phase information caused by chest wall motions. According to Formula (3), the phase change caused by heartbeat and respiration can be expressed as follows:(4)$$\Delta \varphi_{b}=\frac{4\pi }{\lambda }\Delta R,$$

According to Formula (4), the phase modulation degree of IF signal output by mixing is proportional to the motion amplitude of the object [15]. By demodulating the phase, a signal proportional to the displacement of the chest can be directly obtained, which contains the breathing and heartbeat signals of the human body.

### 2.2. 120 GHz Millimeter Radar Front End

Vital signs monitoring radar sensor is featured with small size, low cost and high FMCW linearity. To satisfy these requirements, the 120 GHz integrated millimeter wave radar chip TRX_120 produced by the Silicon Radar Company is adopted as the radio frequency (RF) front end in this paper; a phase-locked loop (PLL) circuit composed of a frequency synthesizer chip ADF4159 and ARM core microcontroller STM32 is used to control millimeter wave voltage controlled oscillator (VCO) of the radar chip, thus achieving a radar front-end with a sweep bandwidth of 120–125 GHz and low power consumption (transmit power −3 dBm). Although the transmit power of the radar antenna is only −3 dBm, according to the radar equation and the reflection characteristics of human skin, the test distance of 1.5 m is much shorter than the unambiguous range of the radar, so the radar transmit power can meet the measurement requirements of human vital signs. The hardware structure of this vital signs monitoring radar is shown in Figure 1.

The IF signal of TRX_120 chip adopts the form of differential output, and when the measuring distance is within the range of 1.5 m, the frequency is dozens of kHz and the voltage amplitude maintains at mV level. Therefore, we set the sampling rate at 200 kHz, selected the low noise operational amplifier chip to form the differential amplifier circuit and the same direction proportional amplifier circuit to realize the amplification and filtering of the differential signal, so as to meet the input signal conditions of the data acquisition card, and make full use of the 16 bit sampling bits. The picture of the 120 GHz radar sensor is shown in Figure 2, and the size of the radar is 80 mm × 80 mm.

Once the radar is powered on, the IF signal passes through the direct-current (DC) isolation capacitor, low noise amplifier circuit and high pass filter, and is then collected and stored by AD and uploaded to the computer. The statistical results show that the normal frequency spectrum of a heartbeat signal is below 3 Hz [16], while the frequency of a respiratory signal is below 0.67 Hz. For FMCW radar, each chirp can obtain the phase information of a corresponding range unit after being processed. Since the phase is proportional to the displacement caused by respiration and heartbeat, each chirp can be regarded as a sampling of heartbeat and respiration signals. According to the Nyquist sampling theorem, the sampling frequency should be more than twice the maximum frequency of heartbeat and respiration so as to keep the oscillation information of respiration and heartbeat completely. Therefore, the frame period of each FMCW pulse was set as 5.6 ms and the duration of frequency ramp was 1.2 ms. Therefore the sampling frequency was 178 Hz, far greater than 3 Hz (heartbeat 180 times/min), which can meet the acquisition requirements of heartbeat and respiratory signals. The chirp configuration of FMCW radar is shown in Figure 3.

In addition, in order to ensure the accuracy of the measurement results, we needed to keep the human body still during the measuring process. Therefore, it was necessary to measure limb jitter in real time. For limb jitter measurement, due to the large mechanical displacement of body shaking, which is beyond the measurement range of phase change, FMCW radar with large bandwidth can be used to calculate the distance variation to realize the measurement. According to the distance resolution formula of radar, the distance resolution of 5 GHz bandwidth is 3 cm, while the amplitude of limb shaking is normally within several centimeters, so it is reasonable to judge the amplitude of limb shaking according to the change of radar measurement range.

### 2.3. Narrow Beam Lens Antenna

In order to achieve the narrow beam control of the antenna, a millimeter wave antenna lens composed of elliptical convex and cylindrical metal layers was designed using polytetrafluoroethylene material (Er = 2.2) in this paper. The initial size of the millimeter wave lens antenna was obtained by optical simulation and then optimized by electromagnetic simulation. Finally, the specific size of the lens antenna was determined: the overall thickness of the lens was H = 20 mm, the diameter of the front ellipse was D = 29 mm, and the ellipse equation was y^2^/27^2^ + x^2^/(27^2^/2) = 1. The distance between the bottom of the lens and the radar antenna was 15 mm. The far-field pattern of the receiving and transmitting antennas after installing the lens was as follows: the 3 dB beam width of the H-plane pattern was 5.19°and the maximum value of the main lobe was 28.7 dB at 0°; the 3 dB beam width of the E-plane pattern was 5.19°and the maximum value was 28.7 dB at 0°. At a distance of 1 m from the antenna and within a radius of 20 mm, the amplitude unbalance degree of the antenna was ±0.5 dB and the phase imbalance was ±10°. Under this condition, the transmitting beam of the antenna can be approximately linearly polarized with the plane electromagnetic wave in the range of 1 m and 20 cm^2^, which can be used to measure the displacement of points on the human body surface. The antenna and lens installation diagram of chip TRX_120 and the far-field pattern of lens antenna E field are shown in Figure 4a–d.

## 3. Experimental Results and Algorithm Design

The quality of vital signs parameters extraction is affected by many factors. For the non-contact measurement system, the vertical distance *R* between the radar and the chest, the relative angle between the main beam direction of the transmitting antenna and the body surface, and the interference distribution around the test environment are the three most important factors. Considering the practical application requirements of radar sensors, the test environment is a laboratory with narrow space, which is similar to the ward environment.

### 3.1. Measurement of Heart Rate and Respiratory Rate

Eight male volunteers with a height of about 1.75 m and a weight of about 65 kg were selected as test objects. During the test, the person about to be tested sat in the chair at a distance of approximately 1.0 m in front of the radar; the heart position of the chest kept a basically vertical relationship with the main beam direction of the radar-transmitting antenna. The heart rate and respiratory rate were measured in the experiment. At the same time, the electrocardiogram (ECG), photoplethysmographic (PPG) pulse wave and respiratory signals measured by the JRTYL-GT6800-10 cardiovascular monitor were compared to verify the correctness of the results. Figure 5 shows the schematic diagram of the respiratory rate and heart rate measurement experiment.

The measurement system samples the in-phase (I) and quadrature (Q) IF signals through the acquisition card and uploads them to the computer via universal serial bus (USB), after which the IF digital signal I + j × Q corresponding to each FMCW pulse is FFT transformed to obtain the spectrum of the IF signal (range FFT), as shown in Figure 6.

It can be seen that the IF signals obtained by using narrow beam antenna have no clutter interference. By selecting the spectrum peak corresponding to the human target we are concerned about, and then calculating the phase θ by using the real part and imaginary part of the frequency corresponding to the peak point of the spectrum, we can thus obtain a phase sequence θ(*n*) (*n* = 1, 2…, N) corresponding to the FMCW pulse. The chest displacement caused by breathing is sometimes greater than $\lambda_{\max}/4$, so the phase sequence has a jump problem. For each pair of adjacent phase values (θ(m),θ(m + 1)), if θ(m + 1) − θ(m) > 180°, it indicates that θ(m + 1) > 0 and θ(m) < 0, θ(m + 1) = θ(m + 1) − 360°; if θ(m + 1) − θ(m) < −180°, which indicates that θ(m + 1) < 0, θ(m) > 0 and θ(m + 1) = θ(m + 1) + 360°. The phase jump process is shown in Figure 7, where the direction indicated by the green arrow is the correct direction in which the phase angle θ changes [17].

After the phase sequence is compensated by the jump, Figure 8a shows the phase sequence changing with time. It can be seen that the phase sequence is a composite signal composed of heartbeat signal and respiratory signal, and the respiratory signal is much stronger than the heartbeat signal. From the demodulated signal, it is easy to read out the respiratory waveform because its peak and trough are very obvious, but it is impossible to observe every heartbeat signal clearly. Due to uncontrollable factors such as system noise in radar itself, the signal quality changes greatly and is unreliable. In order to obtain the respiratory and heartbeat frequencies, we first applied FFT to the phase data. The spectrum is shown in Figure 8b. The five significant peaks represent different meanings respectively. The frequency of the highest peak is 0.26 Hz, which represents the respiratory rate. The third peak represents the second harmonic of respiratory wave, and the last peak of 1.23 Hz represents the heart rate. The first peak represents the frequency of phase sequence data bias line change caused by different breath depth, which is 0.05 Hz. We can see this from the time–domain waveform in Figure 8a. In addition, the spectrum also includes the third and fourth harmonics of respiration and other clutter spectra. Sometimes their amplitude is close to the heart rate, which easily interferes with the judgment of heart rate. From the comparison of the FFT calculation value and the results of JRTYL-GT6800-10, it can be seen that the clutter has a cumulative effect with the increase in acquisition time and merely by using the FFT algorithm to evaluate heart rate, the probability of identifying respiratory harmonic or clutter spectrum value as the heart rate is high, which hinders the practical process of vital signs monitoring radar.

Through analyzing the phase sequence data, we found that the time domain waveform of the respiratory signal was relatively clear, and that there was a big difference in time length between different respiratory cycles. The frequency obtained by the FFT algorithm cannot fully reflect the characteristics of respiratory signals. Therefore, we used a Gaussian low-pass smoothing filter to extract the respiratory signal, and then determined the duration of each breath by looking for the waveform peak in the time domain. The filtered respiratory signal waveform is shown in Figure 9a. Taking the time difference between adjacent peaks in the waveform Δt as abscissa and 60/Δt as ordinate, the curve of respiratory rate (times/min) changing with time is drawn as shown in Figure 9b. Compared with the respiratory rate displayed by the JRTYL-GT6800-10 cardiovascular monitor, the error was within ±2 times/min except for individual scatter points, which was caused by the difference of the time domain waveform feature extraction standards between the two instruments.

The respiratory rate test results of eight volunteers are shown in Table 1, which represents the average ± standard deviation of frequency. According to reference [7], the accuracy *p* of respiratory rate measurement results is defined as the percentage of time when the error between the detected respiratory rate and the reference frequency is within the range of ±10%. The results showed that the accuracy rate of all subjects was above 90%.

### 3.2. Average Heart Rate Measurement

From the time domain waveform of heartbeat signal, it can be seen that it is a kind of non-stationary quasi-periodic signal, and the spectrum of respiratory harmonic and clutter sometimes fall into the band range of heartbeat [18]. Additionally, the signal strength is greater than the fundamental frequency signal of heartbeat signal. Therefore, we used an adaptive notch filter (ANF) to remove respiratory harmonics to improve the accuracy of heart rate detection. The basic principle of an adaptive notch filter is to take the orthogonal signal of a certain center frequency as the reference signal, use the linear combination of the orthogonal signal to track the input signal, and continuously adjust the weight coefficient of the linear combination through the residual error of each step, thus separating the linear correlation part of the input signal with the reference signal and achieving the effect of a narrow band notch. Compared with the conventional digital filter, the ANF has narrow stop band and fast attenuation in band. With two outputs: filter output (y output) and notch output (e output), this device can realize narrow band notch. When the interference or useful signal is a single frequency signal, the adaptive notch filter has good filtering effect [19]. In this paper, the basic principle of the ANF, the structure of which is shown in Figure 10, is to take the orthogonal signal of respiratory harmonic frequency as the reference signal, use the linear combination of the orthogonal signal to track the input signal, and continuously adjust the weight coefficient of the linear combination through the residual error of each step, thus separating the linear correlation part of the input signal with the reference signal and achieving the effect of filtering respiratory harmonic.

Since the amplitudes of the second and third harmonics of the respiratory signal were large, especially the third harmonic, whose amplitude and frequency were close to those of the heartbeat signal, we obtained the respiratory signal frequency f_RR_ through the spectrum peak, by using cos(2π × 2f_RR_t) and cos(2π × 3f_RR_t) as the initial signals, and then using the least mean square (LMS) adaptive iteration to approach the harmonic signal. After filtering out the harmonics, the heartbeat signal was converted to the frequency domain, and the heartbeat frequency was obtained by searching for the peak value in the frequency domain. In order to evaluate the accuracy of the harmonic cancellation algorithm based on the adaptive notch filter, the Doppler radar detection device and photoelectric pulse sensor designed above were used to simultaneously collect human heartbeat information. Additionally, the experimental results were accurately calculated through the pulse wave signal. The pulse sensor can obtain the pulse waveform by collecting the changing information of light transmittance at the end of the finger. This waveform is of high quality and it is convenient to extract the heart rate information, as a consequence of which the heart rate extraction result can be used as the heart rate reference. To reduce the cumulative effect caused by the increase in clutter time length, the short-term phase data of T = 4096 × 0.0056 s, i.e., 22.94 s, were taken as a calculation unit, and the phase signal was filtered by the adaptive notch filter each time it moved forward for 1 s. Additionally, the frequency corresponding to the highest peak point was found in the heart rate range of the filtered FFT spectrum according to some prior knowledge. The upper part of Figure 11 shows the FFT spectrum of the phase signal without the adaptive notch filter processing. It was observed that the second and third harmonic amplitudes of respiration were high. Nevertheless, the peak of the heartbeat spectrum lies between two larger respiratory harmonic peaks. If the peak method is used to estimate the heart rate, the estimated heart rate is 0.78 Hz, which is actually the third harmonic frequency of breathing. The lower part of Figure 11 shows the FFT spectrum of the phase signal processed by the adaptive notch filter, from which it can be seen that the harmonic of the respiratory signal was suppressed, the frequency value of heartbeat signal was significantly higher than that of the respiratory harmonic, and thus the accuracy of heart rate detection was improved. As the heart rate of normal people is between 45–180 bpm, that is, the frequency is between 0.75–3 Hz, the peak value of a respiratory wave below 0.7 Hz is not within our detection range and will not affect the monitoring accuracy of heart rate.

As the phase sampling rate was 178 Hz, the FFT frequency resolution of 4096 sampling points was 0.0435 Hz. When the frequency of signal was not a positive integer multiple of FFT frequency resolution Δf, the spectrum leakage caused by the FFT “fence effect” greatly reduced the accuracy of frequency estimation. For example, for a heart rate assessment in 60 s, the maximum error could be 1.3 bpm. Therefore, we modified the heart rate obtained by the FFT by the adjacent element proportion of spectrum peak method [20]. The specific method is shown in reference [20]. Finally, the heart rate obtained was as follows:(5)$${\mathrm{HR}}_{\mathrm{radar}}=(k_{0}+1-\frac{2\left|X(k_{0}-1)\right|}{\left|X(k_{0}+1)\right|+\left|X(k_{0}-1)\right|})\Delta f,$$
where *k*_0_ represents the frequency sequence number corresponding to the peak value of center rate of the FFT spectrum, and *X*(*k*_0_ − 1) and *X*(*k*_0_ + 1), respectively, represent the amplitude of the left and right sides of the heart rate peak of the FFT spectrum. Considering the standard of actual heart rate measurements, when the difference between the heart rate results obtained by the Doppler radar and the pulse signal reference are less than or equal to 2 bpm (beats per minute), the heart rate extraction result can be confirmed to be correct. At the same time, the beat error rate per minute is taken as the accuracy index of the method. The error rate is expressed as follows:(6)$$\mathrm{Error}\;\mathrm{rate}=\frac{\left|{\mathrm{HR}}_{\mathrm{ref}}-{\mathrm{HR}}_{\mathrm{radar}}\right|}{{\mathrm{HR}}_{\mathrm{ref}}}\times 100(\%),$$
where HR_ref_ is the heart rate obtained from the number of peaks in the pulse wave, and HR_radar_ is the heart rate obtained from the raw phase data through the ANF and FFT algorithm and frequency correction by using radar sensor. In Figure 12, after being processed by an extraction algorithm with the Doppler radar signal and PPG reference signal, respectively, the heart rate detection results of a volunteer whose heart beats too fast were compared, where the heart rate average was 22.94 s. It can be seen that although the heart rate differed from one subject to another, this method can detect the heart rate more accurately.

The heart rate test results of eight volunteers were continuously counted as shown in Table 2, which represented the average ± standard deviation of frequency. The statistical results show that the accuracy rate of all subjects is higher than 90.54%.

### 3.3. Short-Term Heart Rate Estimation

As heart rate changes continuously with time in a certain range, real-time heart rate measurement plays an important role in the evaluation of arrhythmia and other pathological states. Although the improved FFT algorithm can estimate the average heart rate with long-time phase data, for the raw phase sequence data with low signal-to-noise ratio, short-time frequency estimation by means of FFT algorithm will face the problem of a high error rate of heart rate measurement results. According to the analyses in Section 3.2, the raw phase sequence data were mainly composed of the respiratory signal, heartbeat signal, radar system noise and environmental clutter. As a narrow beam lens antenna was used in biological radar, the proportion of environmental clutter was low and thus can be ignored. Therefore, when extracting the heartbeat signal, we only needed to separate and remove the respiratory signal and radar system noise. Since the frequency range of respiratory harmonics and part of the noise coincide with that of the heartbeat signal, it was difficult to remove the noise effectively by means of traditional filtering methods. A Fast-ICA algorithm can well estimate the original signals which were statistically independent and mixed by unknown factors from the observed signals [21]. Therefore, we used the method based on the combination of CEEMDAN and Fast-ICA algorithm to separate the heartbeat signal and evaluate the short-term heart rate in the time domain.

From the raw phase sequence data in Figure 8, we can see that the waveform contour of the respiratory signal is obvious, so we use the method of taking the middle line of the upper and lower envelope lines of the raw phase sequence data waveform to remove the respiratory fundamental wave. Figure 13 is the schematic diagram of obtaining the raw phase sequence data envelope.

By subtracting the middle line of the upper and lower envelope from the raw phase sequence data, the phase signal mainly composed of the heartbeat signal, radar system noise and residual respiratory harmonics can be obtained, whose signal waveform is shown in Figure 14.

Since the preprocessed heartbeat signal with noise is a single channel signal and the Fast-ICA algorithm cannot solve the underdetermined problem, that is, the number of input signals must be greater than or equal to the number of sources, we first adaptively decomposed the signal into multi-dimensional intrinsic mode function (IMF) to meet the requirements of blind source separation (BSS) for positive or over-determined signals. Then, independent component analysis (ICA) was performed on the reconstructed IMF to extract the heartbeat signal. As an improved method for empirical mode decomposition (EMD) and ensemble empirical mode decomposition (EEMD), on the basis of EEMD, CEEMDAN overcomes the problem that EEMD lost with the integrity of decomposition and produces a reconstruction error by adding white noise adaptively. At the same time, CEEMDAN makes it possible for signals to be evenly distributed in the whole frequency band and have continuity in different scales, thus reducing the modal aliasing effect [22]. From the decomposition results of CEEMDAN, it can be seen that the phase sequence data after removing the respiratory fundamental wave can be decomposed into about 15 IMFs. An FIR band-pass filter with a pass-band of 0.8–3 Hz was used to filter the raw phase sequence data to obtain the heartbeat signal with clutter. A correlation calculation can then be conducted with the heartbeat signal and those IMFs, respectively, and then the IMF with the largest correlation coefficient can be identified as the heartbeat signal. However, from the waveform of the heartbeat signal, it can be found that there still exists mode aliasing, which mainly occurs between adjacent IMFs. In order to suppress the mode aliasing effect and improve the accuracy of heart rate monitoring, the IMF signals decomposed by CEEMDAN were divided into three groups. More specifically, the heartbeat signal and its two adjacent IMFs were divided into one group; the heartbeat signal was a single channel signal; the adjacent IMF whose frequency was higher than that of the heartbeat signal was treated as a channel signal; the adjacent IMF whose frequency was lower than that of the heartbeat signal was treated as a channel signal; those IMFs whose frequencies were higher than that of the IMF adjacent to the heartbeat signal added up to form noise; those IMFs whose frequencies were lower than that of the IMF adjacent to the heartbeat signal added up to form other low-frequency clutter. Then Fast-ICA was performed on the three channels according to different weights to obtain the heartbeat signal with the suppressed mode aliasing effect. The classification results after CEEMDAN and Fast-ICA are shown in Figure 15. It can be seen that the combination of CEEMDAN and Fast-ICA can effectively separate the heartbeat signal and radar system noise.

As the separated heartbeat signal has the problem of an unstable waveform, it is easy to make mistakes when directly searching the peak in the time domain. Therefore, the sinusoidal curve fitting method was adopted to reconstruct the heartbeat signal and thus improve the accuracy of the heart rate. The short-term heart rate can be worked out by conducting the sinusoidal curve fitting to the heart signal obtained after separation by taking every 3 s as a segment and finding the peak value of the fitting curve. The curve fitting results of the 3-s heartbeat signal are shown in Figure 16.

We detected the peak value of the heartbeat signal after curve fitting and then calculated the heart rate. Figure 17 shows the short-term heart rate detection results of a volunteer, from which it can be seen that the heart rate measurement results obtained by means of the above algorithm can reflect how the heart rate changes with time.

Compared with the traditional FIR-type band-pass filter, the CEEMDAN and Fast-ICA algorithm have stronger adaptability, and the time–domain waveform of the heartbeat signal was also more stable, which helps to improve the accuracy of short-term heart rate monitoring. However, its shortcomings are also obvious, and its computational complexity is much higher than that of a band-pass filter. According to the short-term heart rate measurement results of five volunteers, the accuracy of the measurement error within ±2 bpm was 87.88%, 84.85%, 87.88%, 84.85%, 90.91%, 81.82%, 84.85%, 81.82%. These experimental results verify the effectiveness of the short-term heart rate extraction algorithm, and the algorithm can track the changing heart rate. However, due to the large amount of calculations, the short-term heart rate measurement results have a time delay of about 100 s. In the whole process of the algorithm, the delay of CEEMDAN and Fast-ICA play a leading role, which determines the delay of the whole algorithm; while other computing time can be ignored for the total delay.

## 4. Conclusions

This paper presents a non-contact vital signs monitoring system based on narrow beam millimeter wave radar and the radar sensor system was implemented and integrated on printed circuit board (PCB). This device possesses the following advantages: (1) the hardware structure of the radar system is compact; (2) millimeter wave lens antenna is used in the front end of radar to realize narrow beam control, which has a strong anti-jamming ability, high SCR of IF signal, and good acquisition performance of displacement of chest position points; (3) the mixer circuit is an orthogonal topology without zero point problems. In this paper, on the basis of the traditional FFT frequency acquisition method, the adaptive notch filter was used to filter out the respiratory harmonics in the spectrum of the heartbeat signal, frequency correction was then conducted and the heart rate was evaluated. This algorithm avoids the interference of respiratory harmonics in FFT algorithm on heart rate, and improves the accuracy of heart rate monitoring. Human experiment results show that the sensor can provide respiratory rate and extract heart rate effectively. At the same time, this paper also realizes the measurement of short-term heart rate, which is beneficial for the analysis of abnormal physiological phenomena such as arrhythmia. For non-professional monitoring and recording, this measurement system has a good practical prospect in terms of comfort. As a non-contact vital signs monitoring device, radar sensor may be used in sleep monitoring, family health care or emergency medical applications. Given that this paper mainly discusses the measurement technology of respiration and heartbeat frequency, it is only limited to measuring the frequency when the human body is still and there is no obvious external mechanical movements in the chest, and no discussion has been made on the actual problems such as dynamic motion compensation. Under the present conditions, it is hard for us to obtain reliable results by means of the proposed method when the human body is in normal motion. When considering how to deal with the phase modulation of the cross range unit caused by the mechanical movement of the body, the improvement of the signal processing method may help to solve this kind of problem and obtain the measurement results in real time. These will be the focus of future work.

## Figures and Tables

**Figure 1 sensors-21-02732-f001:**
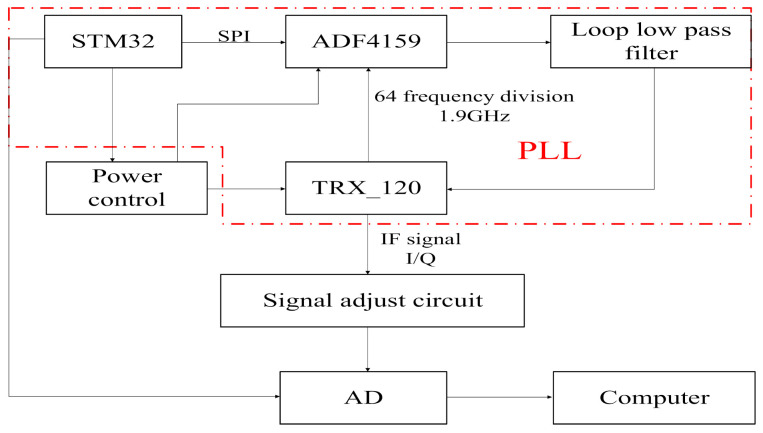
Hardware structure of vital signs monitoring radar.

**Figure 2 sensors-21-02732-f002:**
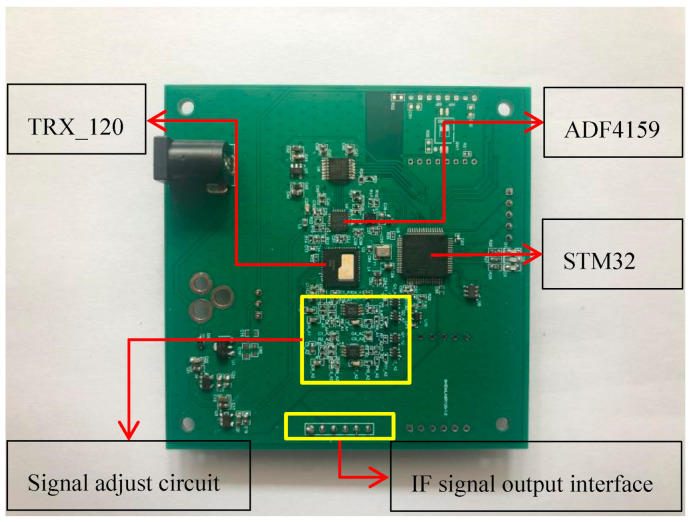
Picture of the 120 GHz radar sensor.

**Figure 3 sensors-21-02732-f003:**
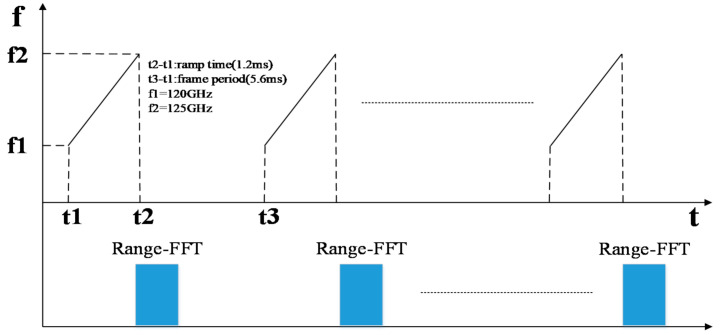
Chirp configuration of FMCW Radar.

**Figure 4 sensors-21-02732-f004:**
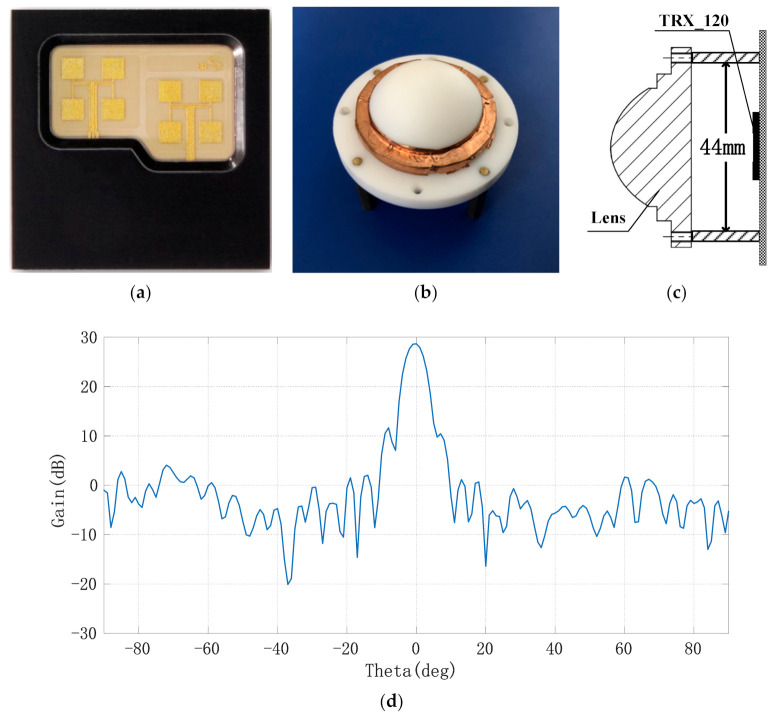
Millimeter wave lens antenna and its far-field pattern; (**a**) layout of millimeter wave radar’s transmitting and receiving antennae; (**b**) picture of the lens; (**c**) assembly diagram of narrow beam lens; (**d**) far-field pattern of lens antenna.

**Figure 5 sensors-21-02732-f005:**
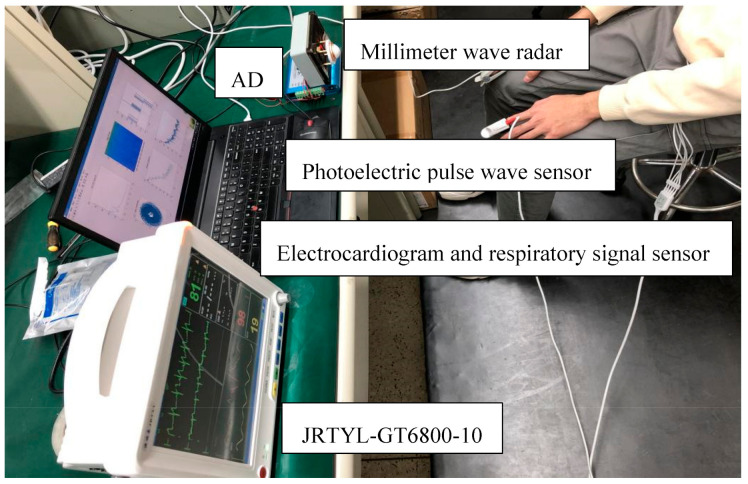
Measurement of respiratory rate and heart rate.

**Figure 6 sensors-21-02732-f006:**
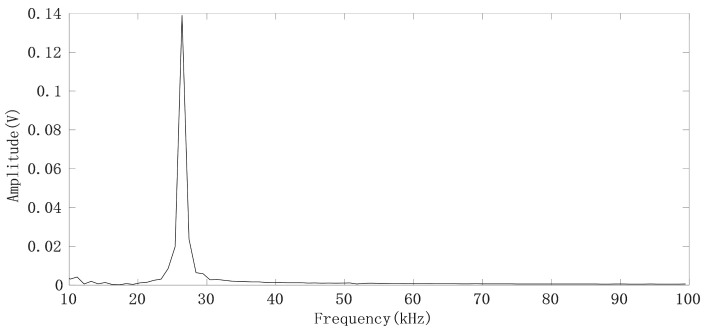
FFT spectrum of the IF signal generated by the FMCW pulse.

**Figure 7 sensors-21-02732-f007:**
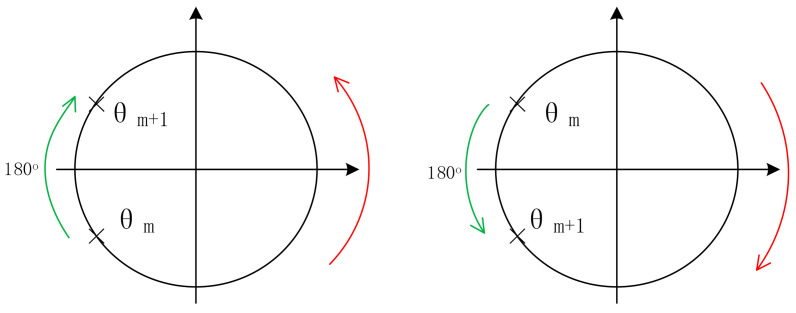
Schematic diagram of the phase jump process.

**Figure 8 sensors-21-02732-f008:**
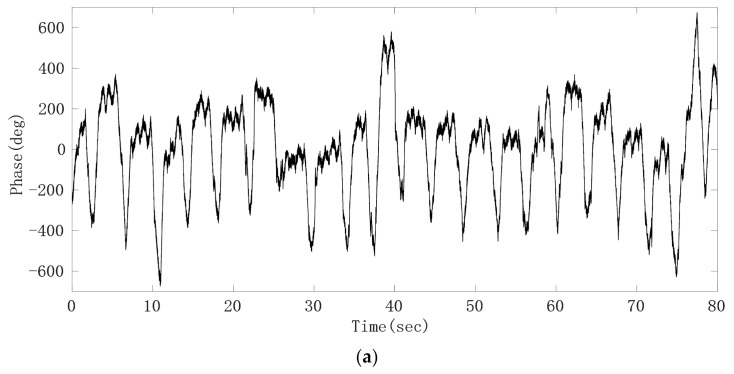
Experimental results of the target; (**a**) raw phase sequence data; (**b**) FFT spectrum of raw phase sequence data.

**Figure 9 sensors-21-02732-f009:**
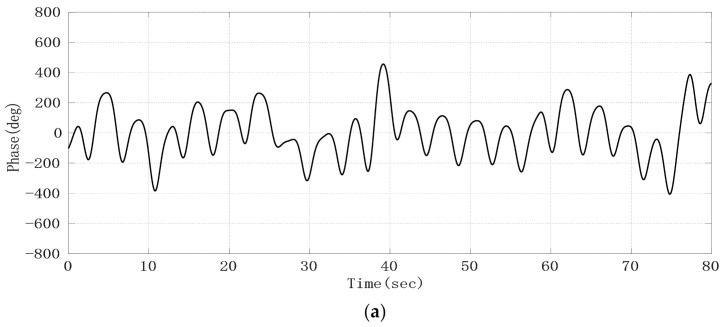
Experimental results; (**a**) respiratory signal after the Gaussian low-pass smoothing filter; (**b**) curve of respiratory rate changing with time.

**Figure 10 sensors-21-02732-f010:**
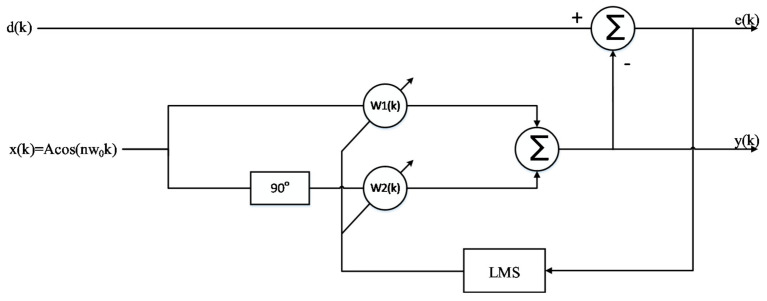
Single frequency adaptive notch filter based on the least mean square (LMS).

**Figure 11 sensors-21-02732-f011:**
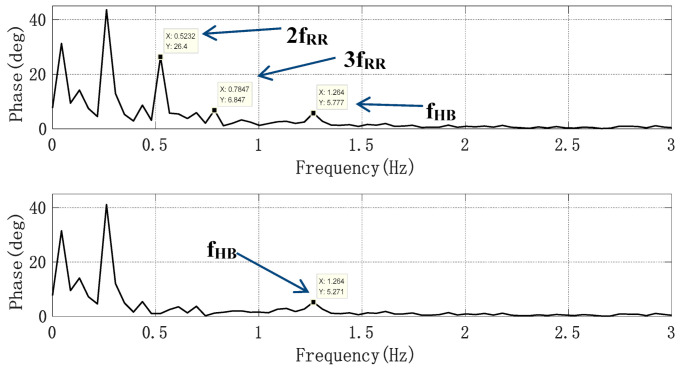
Spectrum without and with an adaptive notch filter (ANF).

**Figure 12 sensors-21-02732-f012:**
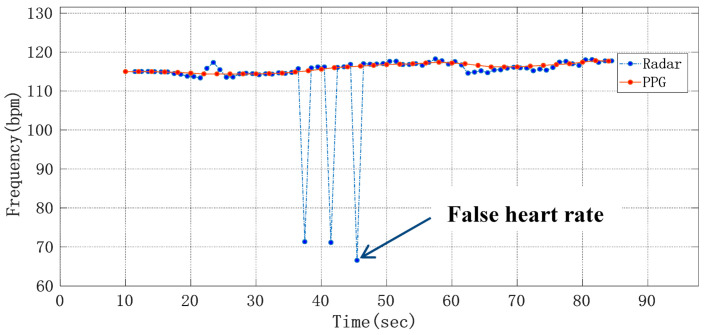
Curve of average heart rate changing with time.

**Figure 13 sensors-21-02732-f013:**
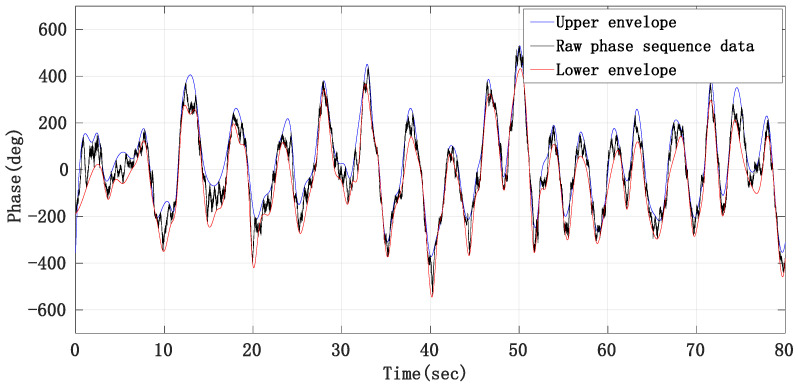
Schematic diagram of eliminating respiratory fundamental wave by envelope method.

**Figure 14 sensors-21-02732-f014:**
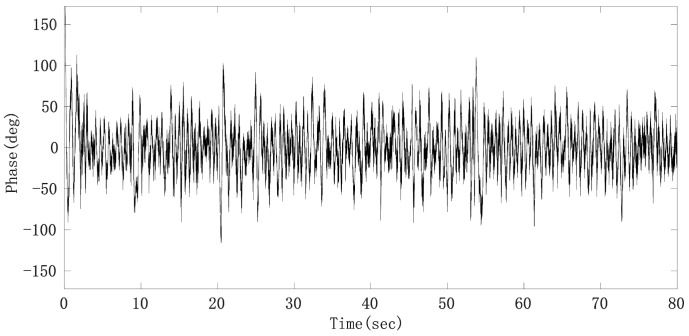
Phase sequence data after removing respiratory fundamental wave.

**Figure 15 sensors-21-02732-f015:**
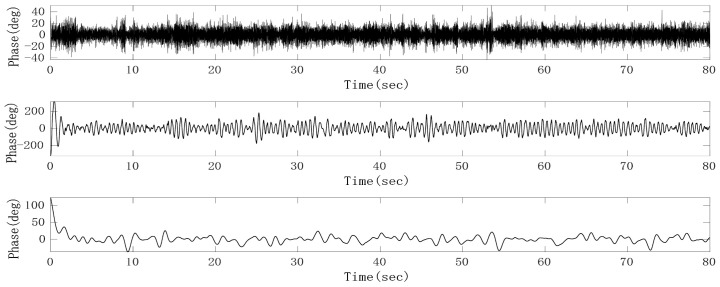
Heartbeat signal (**middle**), noise (**upper**) and other clutter (**lower**) separated by complete ensemble empirical mode decomposition with adaptive noise (CEEMDAN) and fast independent component analysis (Fast-ICA).

**Figure 16 sensors-21-02732-f016:**
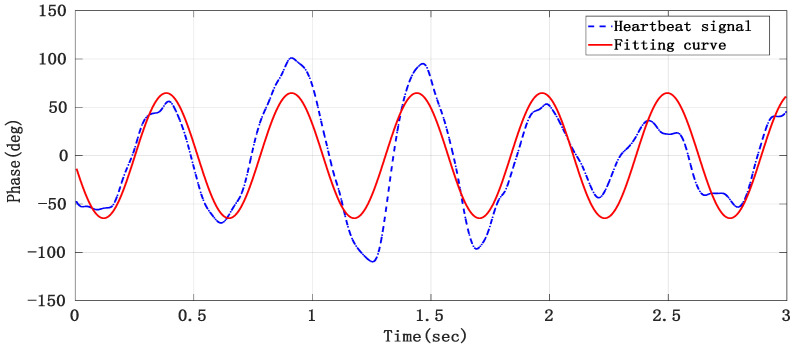
The heartbeat signal obtained after curve fitting.

**Figure 17 sensors-21-02732-f017:**
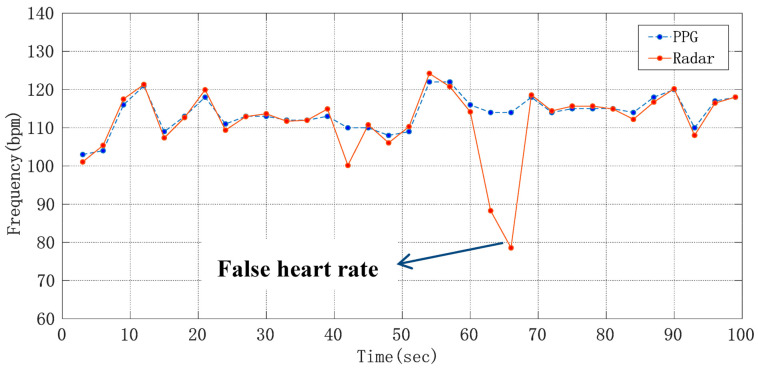
Diagram about how short-term heart rate changes with time.

**Table 1 sensors-21-02732-t001:** Measurement results of the average respiratory rate.

Subject	AVG.RR (times/min)		ACCURACY (%)
	RADAR	JRTYL-GT6800-10	*p* = 10%
1	16.3742 ± 2.2800	15.9412 ± 1.7489	90.48%
2	15.6292 ± 1.1695	15.5000 ± 0.7303	94.74%
3	21.4651 ± 1.3285	21.0000 ± 0.8944	92.31%
4	21.3557 ± 1.7329	21.0625 ± 1.3889	92.59%
5	17.4923 ± 2.4528	17.6250 ± 1.7464	95.45%
6	22.8918 ± 2.7344	22.5263 ± 2.1952	94.12%
7	15.1989 ± 3.8145	14.6667 ± 3.5810	90.91%
8	24.7869 ± 2.2697	24.4737 ± 1.9255	97.30%

**Table 2 sensors-21-02732-t002:** Average heart rate measurement results.

Subject	AVG.HR (bmp)		ACCURACY (%)
	RADAR	PPG	Error Rate ≤ 3%
1	68.4005 ± 1.7972	68.4263 ± 1.7729	94.59%
2	65.8322 ± 1.6590	65.8474 ± 1.2203	93.24%
3	82.8932 ± 1.6739	82.8821 ± 1.5189	100.00%
4	69.8220 ± 2.4188	70.0237 ± 2.1625	90.54%
5	115.8871 ± 1.3015	115.9395 ± 1.1159	93.24%
6	75.0536 ± 2.4188	74.8667 ± 2.1336	91.89%
7	84.0917 ± 1.3294	84.6667 ± 0.8165	94.59%
8	79.6737 ± 1.7707	78.8824 ± 1.7987	91.89%

## Data Availability

The data that support the findings of this study are available on request from the corresponding author. The data are not publicly available due to their containing information that could compromise the privacy of research participants.

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
