# Peer review of "Non-Contact Monitoring of Human Vital Signs Using FMCW Millimeter Wave Radar in the 120 GHz Band"

_sensors, 2021, doi:10.3390/s21082732_

Round 1

Reviewer 1 Report

FMCW 120GHz milimeter wave was used to detect heart beat and respiratory rate.

A few notes about this manuscript are as follows.

in line 13: lens, maybe.

equations (1) and (2) seem broken.

in line 150: add discussions on the distance between radar tx and a person versus the tx power.

add the real measurement pcb boards (or systems) showing every details of the components of Figure 1. Figure 4 is not enough.

Author Response

Dear reviewer:

Thank you for your comments on our manuscript entitled “Non-contact Monitoring of Human Vital Signs Using FMCW Millimeter Wave Radar in the 120GHz Band” (ID:sensors-1154399). Those comments are very helpful for revising and improving our paper. We have studied the comments carefully and made corrections which we hope meet with approval. The main corrections are in the manuscript and the responds to the reviewer’ comments are as follows.

Comments and Suggestions for Authors:FMCW 120GHz milimeter wave was used to detect heart beat and respiratory rate.

A few notes about this manuscript are as follows.

  1. In line 13: lens, maybe.

Response:Yes, it's a spelling mistake. We've changed the word to lens. (In red)

  1. Equations (1) and (2) seem broken.

Response:We found that when we converted the word.docx of the manuscript to PDF format, Equations (1) and (2) seem broken. So we used the formula editor to edit the formula again.

Before revision:

After revision:

  1. In line 150: add discussions on the distance between radar tx and a person versus the tx power.

Response:We have added the discussion of radar transmit power and measured distance in lines 152-155.

  1. Add the real measurement pcb boards (or systems) showing every details of the components of Figure 1. Figure 4 is not enough.

Response:We have added the real measurement pcb boards. Specifically, they include the radar front-end pcb board (Figure 2) and AD acquisition card HK_USB6202-S V1.1 (Figure 5). A phase-locked loop (PLL) circuit composed of frequency synthesizer chip ADF4159 and ARM core microcontroller STM32 is used to control millimeter wave voltage controlled oscillator (VCO) of radar chip, thus achieving a radar front-end with a sweep bandwidth of 120-125GHz and low power consumption.

The picture of the radar is shown below:

The picture of the AD acquisition card is shown below:

Once again, thank you very much for your constructive comments and suggestions which would help us both in English and in depth to improve the quality of the paper.

Kind regards,

Corresponding author : Wenjie Lv

E-Mail: [email protected]

Dear reviewer:

Thank you for your comments on our manuscript entitled “Non-contact Monitoring of Human Vital Signs Using FMCW Millimeter Wave Radar in the 120GHz Band” (ID:sensors-1154399). Those comments are very helpful for revising and improving our paper. We have studied the comments carefully and made corrections which we hope meet with approval. The main corrections are in the manuscript and the responds to the reviewer’ comments are as follows.

Comments and Suggestions for Authors:FMCW 120GHz milimeter wave was used to detect heart beat and respiratory rate.

A few notes about this manuscript are as follows.

  1. In line 13: lens, maybe.

Response:Yes, it's a spelling mistake. We've changed the word to lens. (In red)

  1. Equations (1) and (2) seem broken.

Response:We found that when we converted the word.docx of the manuscript to PDF format, Equations (1) and (2) seem broken. So we used the formula editor to edit the formula again.

Before revision:

After revision:

  1. In line 150: add discussions on the distance between radar tx and a person versus the tx power.

Response:We have added the discussion of radar transmit power and measured distance in lines 152-155.

  1. Add the real measurement pcb boards (or systems) showing every details of the components of Figure 1. Figure 4 is not enough.

Response:We have added the real measurement pcb boards. Specifically, they include the radar front-end pcb board (Figure 2) and AD acquisition card HK_USB6202-S V1.1 (Figure 5). A phase-locked loop (PLL) circuit composed of frequency synthesizer chip ADF4159 and ARM core microcontroller STM32 is used to control millimeter wave voltage controlled oscillator (VCO) of radar chip, thus achieving a radar front-end with a sweep bandwidth of 120-125GHz and low power consumption.

The picture of the radar is shown below:

The picture of the AD acquisition card is shown below:

Once again, thank you very much for your constructive comments and suggestions which would help us both in English and in depth to improve the quality of the paper.

Kind regards,

Corresponding author : Wenjie Lv

E-Mail: [email protected]

Reviewer 2 Report

The manuscript presents a non-contact vital signs monitoring system based on narrow beam millimeter wave radar. The method mainly combining CEEMDAN and Fast-ICA is implemented to separate respiration and heartbeat, and realize the short-term heart rate measurement. In generally, the methods used in this paper are commonly, the novelty of theory, techniques, method have space to strengthen. Description and sentence organization need to be improved. Some minor suggestions would like to be provided to improve the manuscript. 1. English needs to be modified and polished. 2. It is mentioned in the paper that the amplitude of limb shaking is normally within several centimeters, and as we know, the amplitude of human breathing is also within a few centimeters. So how to distinguish these two kinds of movements? 3. Some quality of figures need to improve. Fig. 3 and Fig. 4 are not clear. The abscissa of Fig. 7(b) and Fig. 10 do not start from 0 and need to be revised. Please supplement illustration to each subgraph of Fig. 7(a).

Author Response

Dear reviewer:

Thank you for your comments on our manuscript entitled “Non-contact Monitoring of Human Vital Signs Using FMCW Millimeter Wave Radar in the 120GHz Band” (ID:sensors-1154399). Those comments are very helpful for revising and improving our paper. We have studied the comments carefully and made corrections which we hope meet with approval. The main corrections are in the manuscript and the responds to the reviewer’ comments are as follows.

Comments and Suggestions for Authors:The manuscript presents a non-contact vital signs monitoring system based on narrow beam millimeter wave radar. The method mainly combining CEEMDAN and Fast-ICA is implemented to separate respiration and heartbeat, and realize the short-term heart rate measurement. In generally, the methods used in this paper are commonly, the novelty of theory, techniques, method have space to strengthen. Description and sentence organization need to be improved. Some minor suggestions would like to be provided to improve the manuscript.

  1. English needs to be modified and polished.

Response:We have modified the words and grammar of some sentences, as shown in the green words in the manuscript.

  1. It is mentioned in the paper that the amplitude of limb shaking is normally within several centimeters, and as we know, the amplitude of human breathing is also within a few centimeters. So how to distinguish these two kinds of movements?

Response:We use a narrow beam lens antenna to irradiate the position of the thoracic cavity outside the human heart. According to Reference 13, the abdominal displacement caused by breathing can exceed 12mm, but the displacement caused at the position of the thoracic cavity outside the heart is less than 12mm, and its RMS displacement is 2mm. The amplitude of limb shaking is generally a few centimeters, so we can use the radar intermediate frequency signal to calculate the change in the distance of the human body to determine whether the limb shaking has occurred. For a radar with a range resolution of 3cm, if the human body shakes during the measurement process, the phenomenon of crossing the distance unit will occur. The schematic diagram is shown in the figure below:

(1). When the human body does not have obvious limb shaking during the vital signs measurement process, the distance value measured by the radar remains unchanged.

  • . When the human body has obvious limb shaking during the vital signs measurement process, the distance value measured by the radar changes.
  1. Some quality of figures need to improve. Fig. 3 and Fig. 4 are not clear. The abscissa of Fig. 7(b) and Fig. 10 do not start from 0 and need to be revised. Please supplement illustration to each subgraph of Fig. 7(a).

Response:(1) We have replaced Fig. 3 and Fig. 4 (now Fig. 4 and Fig. 5) with higher definition pictures.

Before revision:

After revision:

(2)The abscissa of Fig. 7 (b) and Fig. 10 (now Fig. 8 (b) and Fig. 11) has been revised to start from 0.

Before revision:

After revision: 

(3). The upper figure in Fig.7(a) (now Fig.8 (a)) is the raw phase sequence data after demodulation, and the lower figure in Fig.7(a) is the phase data sequence after small step length Gaussian smoothing filtering on the raw phase sequence data in the upper figure, so that the heartbeat waveform is clearer. Since the small step Gaussian smoothing filter has no effect on the extraction of life signals, we delete the lower figure in Figure 7(a).

Before revision:

After revision:

Finally, we have also made improvements to the method of combining CEEMDAN and FastICA to extract heartbeat signals. See lines 482-500 of the manuscript for details.

Once again, thank you very much for your constructive comments and suggestions which would help us both in English and in depth to improve the quality of the paper.

Kind regards,

Corresponding author : Wenjie Lv

E-Mail: [email protected]

Reviewer 3 Report

This paper presents the non-contact vital sign monitoring using 120 GHz FMCW radar. The experimental results show the effectiveness of the proposed method. However, this paper has some concerns to be addressed as follows.

  1. Although Section 3 consists of the experimental results and algorithm design, these two contents should be described in separate sections. At first, the authors should describe three algorithms, I.e., respiration rate, heart rate, and the fast heart rate estimation algorithms in Section 3, and then show the experimental results in Section 4.
  2. To show the effectiveness of the proposed the short-term heart rate estimation, the authors should discuss the performance comparison between the two proposed heart rate estimation methods.
  3. To validate the proposed method, more subjects are needed in the experiments.
  4. In the short-term heart rate estimation, how long is the time window to estimate the heart rate?
  5. It is explained that IMF signals are divided into 3 groups according to the frequency. What is the definition of the three groups? Also, how the IMF signals are divided into 3 groups?

Author Response

Dear reviewer:

Thank you for your comments on our manuscript entitled “Non-contact Monitoring of Human Vital Signs Using FMCW Millimeter Wave Radar in the 120GHz Band” (ID:sensors-1154399). Those comments are very helpful for revising and improving our paper. We have studied the comments carefully and made corrections which we hope meet with approval. The main corrections are in the manuscript and the responds to the reviewer’ comments are as follows.

Comments and Suggestions for Authors:This paper presents the non-contact vital sign monitoring using 120 GHz FMCW radar. The experimental results show the effectiveness of the proposed method. However, this paper has some concerns to be addressed as follows.

  1. To show the effectiveness of the proposed the short-term heart rate estimation, the authors should discuss the performance comparison between the two proposed heart rate estimation methods.

Response:We have added a comparative discussion in lines 525-529 of the manuscript.

  1. Although Section 3 consists of the experimental results and algorithm design, these two contents should be described in separate sections. At first, the authors should describe three algorithms, I.e., respiration rate, heart rate, and the fast heart rate estimation algorithms in Section 3, and then show the experimental results in Section 4.

Response:If the experimental results and algorithm designs are described in different sections, the logic and hierarchy of the paper will be clearer. Nevertheless, in this way, the consistency of the experimental algorithm and the experimental results may be poor. For manuscripts involving multiple algorithms, it may be more appropriate to arrange the algorithms and the corresponding experimental results in one section. Therefore, we still keep the original section arrangement, and the algorithms are described in more detail in section 3.

3.To validate the proposed method, more subjects are needed in the experiments.

Response:We added three volunteers and made a statistical analysis of the experimental results.

  1. In the short-term heart rate estimation, how long is the time window to estimate the heart rate?

Response:We use the raw phase sequence data with a duration of 100s for short-term heart rate estimation and calculate a heart rate value every 3s to obtain a total of 33 heart rate values.

  1. It is explained that IMF signals are divided into 3 groups according to the frequency. What is the definition of the three groups? Also, how the IMF signals are divided into 3 groups?

Response:From the decomposition results of CEEMDAN, it can be seen that the phase sequence data after removing respiratory fundamental wave can be decomposed into about 15 IMFs. An FIR band-pass filter with a passband of 0.8-3Hz is used to filter the raw phase sequence data to obtain the heartbeat signal with clutter. Conduct correlation calculation with this heartbeat signal and those IMFs respectively, and then the IMF with the largest correlation coefficient is the heartbeat signal. However, from the waveform of heartbeat signal, it can be found that there still exists mode aliasing, which mainly occurs between adjacent IMFs. In order to suppress the mode aliasing effect and improve the accuracy of heart rate monitoring, the IMF signals decomposed by CEEMDAN are divided into three groups. More specifically, the heartbeat signal and its two adjacent IMFs are divided into one group; the heartbeat signal is a single channel signal; the adjacent IMF whose frequency is higher than that of the heartbeat signal is treated as a channel signal; the adjacent IMF whose frequency is lower than that of the heartbeat signal is treated as a channel signal; those IMFs whose frequencies are higher than that of the IMF adjacent to the heartbeat signal will add up to form noise; those IMFs whose frequencies are lower than that of the IMF adjacent to the heartbeat signal will add up to form other low-frequency clutter. Then FastICA is performed on the three channels according to different weights to get the heartbeat signal with suppressed mode aliasing effect.

Once again, thank you very much for your constructive comments and suggestions which would help us both in English and in depth to improve the quality of the paper.

Kind regards,

Corresponding author : Wenjie Lv

E-Mail: [email protected]

Round 2

Reviewer 1 Report

Now all my comments were answered appropriately.

Reviewer 2 Report

All of the comments have been addressed. The current version has been improved.

Reviewer 3 Report

No further suggestions.